# Changing Hydrosocial Cycles in Periurban India

Carsten Butsch [1,*] , Shreya Chakraborty [2], Sharlene L. Gomes [3], Shamita Kumar [4] and Leon M. Hermans [3,5]

1    Department of Geosciences, Institute for Geography, University of Cologne, DE-50923 Cologne, Germany
2    South Asia Consortium for Interdisciplinary Water Resources Studies, Secunderabad 500 094, India;
     shreya@saciwaters.org
3    Faculty of Technology, Policy and Management, Delft University of Technology,
     2628 BX Delft, The Netherlands; S.L.Gomes@tudelft.nl (S.L.G.);
     L.M.Hermans@tudelft.nl or l.hermans@un-ihe.org (L.M.H.)
4    Institute of Environment Education and Research, Bharati Vidyapeeth Deemed University,
     Pune 411 043, India; shamita@bvieer.edu.in
5    Land and Water Management Department, IHE Delft Institute for Water Education,
     2611 AX Delft, The Netherlands
*    Correspondence: butschc@uni-koeln.de; Tel.: +49-221-4704-142

**Abstract:** India's urbanisation results in the physical and societal transformation of the areas surrounding cities. These periurban interfaces are spaces of flows, shaped by an exchange of matter, people and ideas between urban and rural spaces—and currently they are zones in transition. Periurbanisation processes result inter alia in changing water demands and changing relations between water and society. In this paper the concept of the hydrosocial cycle is applied to interpret the transformation of the waterscapes of six periurban villages in the fringe areas of Pune, Hyderabad and Kolkata. In doing so, three specific aspects will be investigated: (1) the institutions shaping the hydro-social cycle, (2) the interplay between water as a livelihood-base and the waterscape, (3) the interplay between the waterscape and water as a consumption good. This approach opens new views on periurban interfaces as emerging mosaic of unique waterscapes. The meaning of water, the rights to access water and the water related infrastructure are constantly renegotiated, as permanently new water demands emerge and new actors enter the scene. Especially this process-based understanding links the theoretical lens of the hydrosocial cycle with the object of investigation, the periurban space.

**Keywords:** periurban; water; livelihoods; institutions; household; hydrosocial cycle; vulnerability; India

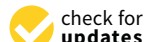

## 1. Introduction

India's current urbanisation process is mutually connected to transformations in various domains. It stimulates societal transformation—while being at the same time driven by this; it results in the physical transformation of spaces—while being dependent on the ecosystem services of the growing urban agglomerations' hinterlands; it changes the economic situation of the country—while being fueled by the forces unleashed following the economic liberalization in 1991. In the next 30 years the number of urban dwellers is expected to double to 877 m [1]. Land use for urban areas is predicted to grow from 25 m hectares in 2005 to 45 m hectares in 2050 [2].

The effects of this powerful process are not limited to the cities proper, but also affect the periurban spaces surrounding these cities. These periurban interfaces are spaces of flows, shaped by an exchange of matter, people and ideas between urban and rural spaces [3]. They are characterized by high heterogeneity of land uses [4,5], actors with often conflicting interests [6] and inadequate governance structures [7–9]. The urban transformation results in an intensification of flows between urban and periurban areas, infrastructural developments connecting the urban and the periurban, the development of settlement structures and the emergence of new economic activities, like industries. Yet,

urbanization processes do not necessarily result in the shift of periurban spaces to urban spaces, but that they can also follow different development pathways within the larger agglomerations and thus deliver for example important ecosystem services [3].

One distinct feature of India's urban transformation is that many developments are taking place in an environment of informality, which is created simultaneously from below and from above [10]. In periurban spaces this is also the result of a lack of adequate governance. While cities are administered by municipal corporations, which are often underfunded, but at least theoretically have access to adequate managerial skills and administrative manpower, peri-urban villages are mostly under rural self-governance, the panchayati raj. These institutions of local self-governance were created following the acceptance of the 73rd constitutional amendment act in 1992, but never became the powerful institution they should be on paper [11]. Thus, periurban areas are often administered without the necessary managerial skills and by weak actors, who often cannot pursue their vision of development against financially potent actors, like developers or industrialists, for whom periurban areas are new sites for their business activities—where local authorities can often be handled easier, because of the power disbalance. In the absence of a strong formal government, the governance of resources takes places by informal institutions—which need to be incorporated in land use and waterscape planning for an adequate water risk management [12,13]. Thus, it would be wrong to equate weak government with lack of governance [14]. But in this situation, periurban areas often appear as an unplanned mosaic of land uses, characterized by sub-standard infrastructure provision [15,16].

In this paper the changing relation of water and society in periurban India will be analysed, based on exploratory field work in six villages. Through the application of a hydrosocial cycle perspective we will contribute to the understanding of the political ecology of these specific zones in transition. Here, a range of actors battles for scarce resources like land, labor—and water. Being a country with high water stress, where a share of 40–80% of the available renewable surface water is withdrawn, the effects of the periurban transformation on water availability are a major concern [17–19]. In the next section, the concept of the hydrosocial cycle will be discussed, as it provides the theoretical lens for our investigation. Then the three areas of in-depth investigation—institutions regulating the access to water, water-based livelihoods and household water—will be introduced. After descriptions of the three metropolitan areas and the methods, the development pathways of six periurban villages will be drafted. In the discussion these six "tales" will be interpreted against the concept of the hydrosocial cycle before the paper ends with a conclusion.

### 1.1. The Periurban Hydrosocial Cycle

Political ecology scholars have been working on the relation between water and society for the last two decades, drawing back on older works, that framed water as a central element of socio-technical systems [20]. From this perspective, water landscapes in a given space at a given time—the waterscapes—are hybrids, which are partly natural and partly social. They are the product of historical developments, power relations and their situatedness in space [21]. The core of the waterscapes concept is, that flows of available water follow the laws of physics in a landscape that is produced by laws of society (institutions). In this waterscape, access to water is regulated by social rules and norms. Important elements of waterscapes are infrastructure elements like dams, pipelines, irrigation systems, which Swyngedouw terms [20]. In this production the access to and the control over the resource is changed and renegotiated. From a political ecology perspective analysis then aims at understanding how environmental degradation, environmental protection and the creation of new environments are the outcome of power struggles [22].

Going beyond the concept of waterscapes, scholars have developed the concept of the hydrosocial cycle as an alternative to the well-known concept of the hydrological cycle [23]. This concept looks at the recursive relationships of water and societies. It not only looks at how societies shape waterscapes, but also considers how societies are shaped by water

and how water is not only subject to social practices but also produces social practices. In this way, the disposition of water and societal organizations affect each other in a cyclical process. Thus, specific institutions, infrastructures and processes emerge in reactive ways, which are highly context specific and not necessarily stable over time, which is of special interest in the periurban context as a zone in transition.

An important aspect of this concept is that water has a very context specific meaning [24] going beyond the substance itself. These specific meanings are produced by the social practices around them. Water can be drinking water, domestic water or irrigation water; it can be holy, if it flows in the river Ganga or it can be dangerous if it cuts off Mumbai's downtown area, as during the 2005 flooding event; it can be ritually impure wastewater, poisonous sewage or the input for wastewater-based agriculture. Thus, "'water' is never simply $H_2O$ but always produced as a particular 'water', materially and discursively, and within specific moments, contexts and relations" [24]. In the case studies, water will be addressed as a base for periurban livelihoods and as a consumption good.

An important aspect of the concept of the hydrosocial cycle, especially in the quickly transforming periurban areas, is that water embodies conflicts between different users [25]. As water crosses political boundaries—in our case from the periurban to the urban to the periurban again (as effluents)—it is subject to power struggles. Indeed, water has also become a substance for empowering local communities under the Integrated Water Resource Management guidelines, which has not always been successful [26].

In periurban areas, different institutional actors exercise power to gain as much water for their interests as possible. Conflicts arise between municipal bodies, trying to secure water for their quickly growing populations, agricultural departments trying to secure water for irrigation and the private actors like industries or real estate developers who need water for their businesses. Periurban areas are zones in transition with power-relations and institutions in flux. The control over water is constantly renegotiated here. In the Indian context periurban farmers suffer from "water grabbing" [27], as they compete over the scarce resource with industries, which are promoted by state governments for achieving economic growth, and specific desires of the growing affluent class, which manifest in high water consumption (e.g., for golf courses). Similar phenomena have been described in other regions of the word, e.g., the "urbanizing" of "rural waters" in Lima [28].

In water scarce cities, like Hyderabad, there is a heavy extraction of water for urban demands in the surrounding periurban areas, which results in a shift of farmers to water vending business—and by that they transform agricultural water (they are allowed to extract) into drinking and domestic water [29]. These new practices then also unfold societal relevance, as they result in a shift of periurban livelihoods—of the farmers themselves directly, but indirectly also e.g., of landless labourers. As these examples highlight, the periurban transformation results in quick and sometimes fundamental alterations of the hydrosocial cycle.

### 1.2. Three Perspectives on Water Transformations in Periurban Areas

The periurban transformation means an increase of flows (resources, people, capital, ideas) between urban, periurban and rural areas [7,30,31]. This is accompanied by an increase of settlement activities, economic activities and often an intensification of agriculture. The relation of water and society is changing substantially with the periurban transformation, putting the resource under pressure: in situ the demand is increasing, where industries settle, new housing structures come up and agriculture is being intensified and the "thirsty cities" [32] not only source water from their periurban surroundings but also release their effluents there [33], often resulting in a depletion of downstream surface and ground water [34–37].

This transformation of periurban waterscapes will be analyzed in what follows from three specific viewpoints: (i) from the institutions shaping the hydro-social cycle, (ii) from a livelihood perspective looking at the interplay between water as a livelihood-base and

the waterscape, (iii) from a household perspective looking at the interplay between the waterscape and water as a consumption good.

### 1.2.1. Institutions

Here, institutions are defined as 'rules' that structure behavior and social interactions [38]. They comprise laws, policies, regulations as well as local practices, customs and traditions. The literature categorizes them in several ways (e.g., formal vs. informal, rules-in-form vs. rules-in-use) [39,40]. In periurban areas, both types of institutions are found to co-exist. Water governance is based on a set of underlying institutions which include rules defining the roles and responsibilities for water supply and regulation; operational rules, quality standards, pricing and other management rules. These institutions offer clarity and guidance, yet there are several reasons why periurban institutions prove ineffective. Typically, institutions are arranged along administrative boundaries. The Nagar Panchayats as an institutional construct specifically created for these zones in transition are not implemented locally [41]. Therefore, periurban areas often lie at the intersection of more than one administrative jurisdiction, leaving them with fragmented, unclear, or overlapping rules [8,42]. Clear definition of roles and responsibilities are challenging in this context as institutions can also be misinterpreted [40]. A mosaic of institutional arrangements emerges in periurban areas, where some villages continue to remain under a rural panchayat system, others become a new municipality or get absorbed by a neighboring municipality. This also determines the utilization of local water resources, e.g., water supply in municipal areas is often piped, treated surface water in contrast to areas under rural development. Often, gaps in formal institutions fosters the emergence of informal institutions through collective action or community-based approaches [43,44] with both types of institutions actually shaping waterscapes. Thus, the changing relation of water and society finds its expression in the alteration of institutions governing water.

Periurban water in specific lacks governance for three reasons [45]: (i) weak regulatory oversight in periurban institutions attract illegal development or polluting activities; (ii) existing water supply institutions are not translated on the ground due to physical and other socio-political constraints [43,46]; (iii) periurban institutions are ill-equipped to manage changing water needs due to increased drinking water demand and emerging livelihoods resulting in an overexploitation of resources [33]. These governance regimes substantially shape the periurban waterscape and affect the two other fields of investigation.

### 1.2.2. Livelihoods

Water-based livelihoods are among the main reasons for alterations in the periurban hydrosocial cycle. Among them are obvious ones—like farming, animal husbandry, fishing and aquaculture—but also less obvious ones such as floriculture, brick making, pottery, laundry services, car wash services or tanning. Among them periurban agriculture so far received most attention in the literature [3]. For farmers, water is a decisive input factor, that is getting under pressure through periurbanisation [47–50]. In regions, where periurban farmers rely on groundwater for irrigation, they compete for the scarce resource and often face the option to 'go deep or quit' [51,52], i.e., to invest in deeper borewells or move out of agriculture. This puts especially the livelihoods of marginal farmers at stake [52–54].

Some respond to this by shifting to wastewater as a source of irrigation, which is a central topic in the literature on water for periurban agriculture. Most authors highlight positive effects, e.g., previously barren land can be cultivated [55], farmers need less fertilizers, and may even experience a yield increase [56–58], it is available round the year [59–61] and it may offer higher profit margins for non-premium products [62].

Wastewater is also used in aquaculture, which may pose health risks, if fish is contaminated with heavy metals or germs [63,64]. Yet it offers an opportunity for farmers to increase their income [65].

Traditional fishing in lakes or rivers has in the past been relevant for socially deprived communities. With the growing pressure on water bodies, these communities find it increasingly difficult to access their traditional fishing grounds, as access to water bodies is reorganized and renegotiated [66,67].

As the periurban spaces transform, new water-based livelihoods emerge, often conflicting with the traditional ones, e.g., polluting small-scale industries like dyeing [68]. Others are related to the needs of the growing urban areas, like brick making [69], or changing lifestyles, like food vending [70].

Water vending as emerging livelihood is specifically related to the periurban transformation. Depending on the locality, periurban water vendors either serve periurban populations or the city or both. These entrepreneurs ensure the water supply in situations when piped water infrastructure is not available [70,71]. Especially for farmers water vending may become a lucrative alternative to their traditional livelihood [29,72,73], while the massive extraction of water affects the local ground water body [72] and may result in drinking water scarcity in the periurban [73].

### 1.2.3. Water as a Consumption Good

Periurban communities in these settings face problems in securing household water access. Falling outside the boundaries of urban water supply networks, the described institutional lacunae and a lack of prioritization results in an inadequate public water network and poor-quality public water supply [68,74–76]. The periurban is characterized by the creation of "infrastructural archipelagoes" and fragmented landscapes wherein only the affluent communities gain secure water access [14,68,77,78]. For the periurban poor basic water access depends on formal and informal private, community, and hybrid institutions [35]. The spectrum of solutions ranges from private individual household groundwater sources, informal tankers, community tanks and lakes, water user associations to sources built through public-private partnerships [19,29,79–82].

Through a critical lens, these diverse periurban water systems have been assessed as enhancing inequalities. Collective systems such as Water Users Associations are affected by unequal power relations creating unequal ownership and participation making most participants nothing more than 'paying customers' [83]. Informal water suppliers and markets are expensive, provide uncertain quality, have poor financial, hydrotechnological, and organizational capacities, and significantly require mobilization of social capital for access by households [84–86]. Thus, benefits of the market accrue to well-of communities only [19,29,87]. In the Indian context, caste and gender categories also shape the unequal distribution of household water [19].

Risks are related to affordability, flexibility of supply conditions, quality, quantity and continuity of supply, distance [33,35,88–94]. Risk management strategies such as water storage or use of multiple water sources are further shaped by socioeconomic status, social and economic networks, capacity, inter- and intra-household power relations [19,88,95–98].

### 1.3. The Three Metropolitan Regions

The project H2OT2S investigates water related periurban transformations in three Indian cities: Pune, Hyderabad and Kolkata. They were chosen, as they are growing metropolitan areas situated in different hydroclimatic areas of India and in three different states—with differing institutional settings. Thus, the governance of the periurban transformation and of water differs between the case studies, allowing for capturing the nature and plurality of waterscapes in periurban areas that emerge under different policy environments in conjunction with the resource base.

Pune is situated in the rain shadow of the Western Ghats in the midst of an intensively used agrarian area at the confluence of the rivers Mula and Mutha [99]. It is located on the Deccan plateau 560 m above sea-level. Especially the agriculture East of the city is relying on irrigation water collected in dams in the Western Ghats. Pune has been an important city politically and culturally since early modern times [100]. Since Indian

independence the city has witnessed an enormous growth: In 1941 the Pune agglomeration had 375,000 inhabitants [101], in 2011 Pune was the ninth largest agglomeration of India and the eighth largest agglomeration and presently (2020) the agglomeration has approximately 6.6 m inhabitants [1]. Pune has been actively promoted as second industrial hub in Maharashtra (besides Bombay/Mumbai). To manage the development of agglomeration, the Government of Maharashtra has recently established the Pune Metropolitan Region Development Authority, coordinating planning processes of two municipal corporations, three cantonment boards, seven municipal councils, 842 villages and 13 census towns [102].

Hyderabad is the capital of Telangana, located at the heart of the Indian peninsula, on a hard rock granitic aquifer and drained by the Musi river. The region is dotted with a cascading system of water bodies that form a significant waterscape [103,104]. Agriculture is largely seasonal and rainfed to a smaller extend groundwater-based, which is very capital intensive because of the hydrological conditions, or downstream Musi wastewater-based [55,105]. Hyderabad has historically been an important seat of power and since independence the capital of Andhra Pradesh and since 2014 of Telangana [106]. Industrial areas were developed since the 1960s and the Silicon Valley since early 2000, resulting in a rapid population growth from 1.2 million in 1961 [103] to 7.7 million in 2011 [107] making it India's sixth largest metropolis [106]. In 2007 the Municipal Corporation of Hyderabad (MCH) was merged with 12 municipalities to form the Greater Hyderabad Municipal Corporation. Since 2008 the Hyderabad Metropolitan Development Authority (HMDA) was formed to coordinate planning among MCH and 55 peripheral administrative units [106].

Kolkata is located in the water abundant Ganges delta with the Hoogly river running North-South parallel to the city on the West [108]. Majority of the city was a wetland area that has been reclaimed over time. The East Kolkata wetland, a RAMSAR protected area is 12,500 ha in size [109]. Kolkata city is part of the Kolkata Metropolitan Area, including 3 municipal corporations (Howrah, Kolkata and Chandan Nagar), 38 municipalities, 77 non-municipal urban towns, 16 out growths and 445 rural villages [110,111] with a population of 14.04 m [112]. The historic city dates back to the 1700s and continues to be an important urban center for trade, commerce, art and culture. The largest growth in population occurred during India's independence 1947 and the Bangladesh Liberation War in 1971, both leading to mass migrations from Bangladesh [108,113]. Primary water sources like rivers and groundwater face pollution problems and seasonal scarcity. The risk of Arsenic contamination in certain areas of the urban agglomeration is also high.

## 2. Materials and Methods

The findings presented in the next section are based on two sources: (i) exploratory field work for the project H2OT2S (For further information please refer to the project's website: http://saciwaters.org/t2speriurban/, accessed on 27 February 2020) and (ii) to a smaller extent based on data collected by the authors in earlier projects. Primary data from these two sources was supplemented by secondary data from the census and the respective local administrations.

The exploratory field work for H2OT2S took place in March 2019. In the period of two weeks, six researchers visited the six study sites—two villages in each of the three agglomerations. The six sites visited are Paud and Uruli Kanchan (periurban Pune), Badai and Hadia (periurban Kolkata), Anajpur and Bowrampet (periurban Hyderabad). The criteria for choosing the six villages include: (i) being located outside the municipal boundaries of the agglomeration's main urban settlement (ii) changes in the livelihoods described in the census data, (iii) changes in the institutional configuration regarding water, (iv) a diversification water sources for. In total eleven criteria were used to screen for potential study sites in the three metropolitan regions, with no study site matching all of them. Yet, the selection will allow for a comparative study of the diversity of water related periurban transformations in India.

During the exploratory field visits, the researchers combined different qualitative methods to collect information about household water, water-based livelihoods and institutions regulating access to water. The mixed methods approach included (i) transect walks with key informants (one transect walk in each village with a key informant), (ii) unstructured, topic focused interviews with local actors (Paud: 2, Uruli Kanchan: 2, Anajpur: 1, Bowrampet: 2, Hadia: 4 Badai: 3) and (iii) narrative interviews with local populations (Paud: 8, Uruli Kanchan: 4, Anajpur: 3, Bowrampet: 3, Badai: 1). All interactions during these field visits were protocolled. Academics, students and field work assistants from the local partner institutions offered introductions to stakeholders and assisted with translation during the transect walks and the interviews. After the field visits detailed reports were compiled by Carsten Butsch for Pune, Shreya Chakraborty for Hyderabad and Sharlene Gomes for Kolkata, containing summaries of the transect walks and interviews. These were analyzed with a qualitative content analysis.

The information from this main field work was supplemented with data from earlier projects of the PIs and secondary data. These are: (i) in Pune, researchers from Bharati Vidyapeeth University (Pune) and from University of Cologne investigated urban developments and developments in the periurban areas since 2005 in six different funded research projects and research oriented teaching, applying both quantitative and qualitative research methods. Two joint field practicals (in 2009 and 2018) explicitly analyzed transformation in Pune's periurban surrounding [99]; (ii) in Kolkata, researchers from TU Delft worked on the *Shifting Grounds* (2014–2018) project led by an international consortium of academic and local partners, which applied mainly qualitative and participatory research methods [114]. During this project, in-depth research into institutions, livelihoods, and hydrogeology of groundwater resources in peri-urban Kolkata was conducted as inputs for supporting institutional transformations for groundwater management. In addition, field research in Kolkata was conducted in 2019 by Aashna Mittal, an MSc student from Delft University of Technology on community based common pool resource management in peri-urban Kolkata [45,115]; (iii) the field insights from Hyderabad have been enriched through SaciWATERs' experience working on periurban water issues in Hyderabad since 2012 through multiple interdisciplinary and consortium based projects, analyzing periurban water security in South Asian cities, the institutional frames for water access, the informal water tanker market, and conflicts and cooperation surrounding periurban water resources. In this research, both quantitative and qualitative methods were applied.

Based on this rich empirical base, the following sections portrays changes in the hydrosocial cycle of our six study villages.

## 3. Results—Tales of Six Periurban Villages

All study sites are located in the periurban, yet, their recent transformations followed very different pathways (just like Paris and London in Dicken's tale of two cities). Being shaped by comparable forces, they exemplify the broad variety of the periurban. Moving geographically from East to West, first the two villages in periurban Pune will be portrayed, before moving to Hyderabad and Kolkata. Each village portrait will start a vignette and then address the components of the periurban hydrosocial cycle from the three perspectives detailed above (Figure 1), starting with livelihoods and then turning to household water, with the relevant institutional influences on both being discussed in parallel. Each "tale" ends with a short reflection on how periurbanisation has altered the hydrosocial cycle.

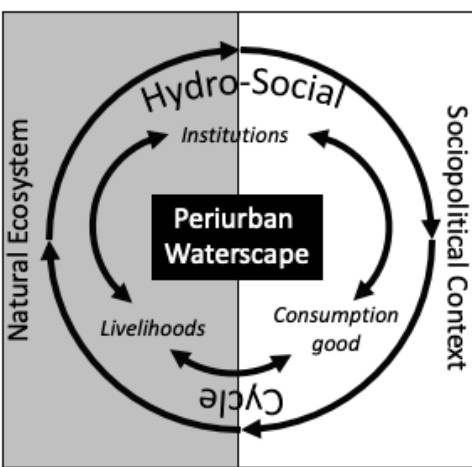

**Figure 1.** A periurban perspective on the hydrosocial cycle (authors' own draft).

*3.1. Periurban Pune—Paud*

With its roughly 4000 inhabitants, Paud is located 30 km West of Pune on a highway. Paud has a small administrative center (second tier government) with a court, a block development office, a hospital etc. The clear separation of different parts of the village is very obvious. The old village core (gram panchayat, market) is located North of the highway, with neighborhoods stratified by caste. South of the highway are newly developed areas with mixed populations, the government buildings and the new township "Playtor". The latter was still under construction, with one apartment block inhabited and four unfinished. The township will eventually have 900 flats. As registered township with the government of Maharashtra, the developers have to guarantee the supply of water and electricity 24 h/7days a week, waste management, health and education facilities. Thus, in Paud there will be two parallel settlements with different rules and regulations, almost equal in size.

There are three traditional water-based livelihoods, which are affected by periurbanisation: fishing, farming, and pottery. Members of the fishing community (Figure 2) highlighted how their livelihoods have changed, as their access to water bodies was restricted. Especially accessing the dams in the Western Ghats has become difficult, as the government decided to auction the right to fish in the dams. As the fishermen cannot pay the same prices as financially potent outsiders, they were excluded from some of their traditional fishing grounds. Now they have to travel longer distances for fishing and their income is stable or decreasing. These institutional changes result in skepticism regarding the sustainability of their livelihood on side of the fishermen.

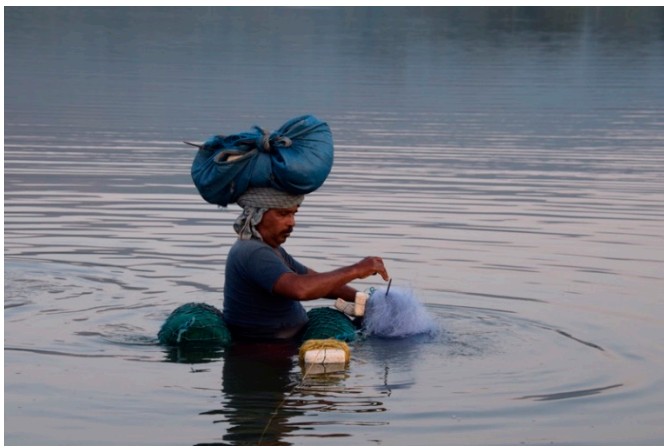

**Figure 2.** Fisherman in Paud (photograph by C. Butsch).

Agriculture in Paud used to be rainfed. In the last years, medium-sized farmers invested in irrigation systems, drawing water from the river Mula to their fields. Only rich farmers could afford the initial investment, which rewards them with higher profits, as agriculture is possible the whole year now. In addition, a dairy cooperative offers a small side business for farmers. Yet, many farmers are convinced, that they are the last generation to farm their ancestors' land. The traditional pottery business has changed, too. Potters increasingly produce for markets in Pune and diversified their products by entering into brick making. At the same time new water-based livelihoods are emerging: car-wash centers and tourism. Many households have members working outside the village, mainly in the industrial cluster in the neighboring village or in Pune.

Paud's domestic water supply is characterized by a multiplicity of water sources. Most households rely on the regular water supply of the local administration. Every household gets 30 min of water daily from a multi-purpose dam. Residents attested a very high quality to this water. The villagers store water in barrels in front of their house or overhead tanks. In some areas of the village, the water pressure in this distribution system is not high enough, therefore some households have a private water connection from the company, which also owns the multi-purpose dam. In general, the domestic water supply is perceived as sufficient by the population.

The hydrosocial cycle in this village is already reconfigured through periurbanisation processes and can be expected to transform even faster in the near future. Traditional occupations are still dominant, but the construction of Playtor, institutional changes and the increasing connectivity to Pune are main drivers of changes in the waterscape. Changing governance patterns, of the fishing grounds for example, will change fishing communities' livelihoods. In the foreseeable future conflicts regarding the access to domestic water can be expected between the inhabitants of the village and of the new township.

*3.2. Periurban Pune—Uruli Kanchan*

In 2011 Uruli Kanchan, 30 km East of Pune, had a population of 30,300 inhabitants [116]. Being still under rural governance, so administrative wise a "village", it is addressed as a "census town". Due to its location on a highway and on one of the major railroad tracks connecting the Mumbai-Pune agglomeration to the East of the country, the village witnessed an enormous growth: the number of inhabitants doubled from 1991 to 2011. If the growth rate remined the same, in 2019 the population of the "village" would then have been almost 42,000 inhabitants. But it is not only the statistics that illustrate Uruli Kanchan's transformation, also the housing structures in place have urban qualities. There are now outside the old village core several gated communities—multi-story walled compounds for affluent citizens.

With respect to livelihoods key informants, inter alia the village development officer, who heads the local administration, described a decline of the primary sector. Only 500 of 15,000 households have an income from farming. In addition, cropping patterns have changed because of infrastructural reconfigurations. In the second half of the 20th century the water availability increased with water being channeled from the Western Ghats, leading to a shift to sugarcane as main crop. Since 2016 water supply for irrigation has been reduced, as a larger share of the fresh water is now used as drinking water for Pune. Farmers in the northern part of Uruli Kanchan now receive grey water for irrigation. A new water-based livelihood related to farming are nurseries (Figure 3), which are situated all along the highway to Uruli Kanchan. These nurseries are high-end, well organized, large sized enterprises with a network of customers all over India. Nurseries can be seen as a follow-up to agriculture, with some also run by outsiders on leased land. Emerging water-based livelihoods are mostly related to the periurban transformation, like car wash facilities, brick making and reverse osmosis (RO) plants of water vendors.

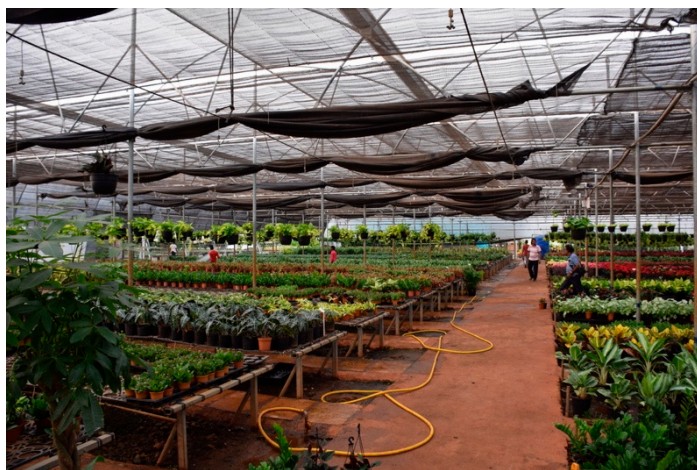

**Figure 3.** Nursery in Uruli Kanchan (photograph by C. Butsch).

RO-plants now, are the main source of drinking water for most households in Uruli Kanchan. One NGO has facilitated the implementation of these RO plants, because of a high burden of gastrointestinal diseases. The NGO set up four RO plants, which will be handed over to the gram panchayat in future. For domestic water, each household has a tap connection outside the house, provided by the gram panchayat. Each household receives 50 L of domestic water per capita per day. The water stems from a dam in the Western Ghats, upstream of Pune. There are plans to install a filtration plant to supply piped high-quality drinking water to each household in future. Respondents from the local administration would like to see Uruli Kanchan to be upgraded to a municipality. This would mean an increase in funds, a change in the administrative structure and consequently an increase of services for the citizens, e.g., regarding the provision of water.

Uruli Kanchan has seen very fundamental transformations of the hydrosocial cycle already. Agriculture has changed, because of changing access to water and the proximity to Pune, which makes it easier to follow alternative livelihoods. Due to the village's high accessibility, farmers make high profits by transforming fields into real estate. Water as a consumption good has changed with the introduction of piped water in every household and the introduction of RO-water for drinking. The latter results in a new set of everyday practices in the households, as water has to be collected/delivered from the RO plants to ensure constant availability of drinking water. This requires new management structures in the households—a mainly taken over by male household members.

### 3.3. Periurban Hyderabad—Anajpur

Located in the Southern part of the Hyderabad agglomeration, Anajpur had 4600 inhabitants in 2011. The village has grown in the decade 2001–2011 by 32.3% [107,116] and is situated outside the agglomerations's Outer Ring Road (ORR), which is attracting urban developments, explaining this fast growth. In addition, Anajpur borders the world's largest film studio, Ramoji Film City, where up to 20 movies are produced in parallel [117]. A textile company was set up in Anajpur in the 1980s. Future developments are clearly visible at the edge of the village where real estate developers have demarcated plots and constructed gravel roads on many former agriculturally used plots.

Several water-related conflicts affected the livelihoods of the local farmers in the past. One open conflictis related to the release of toxic effluents from the textile factory. Local farmers explained that 40 acres of farmland became infertile and poultry farms had to be closed, because the livestock died after consuming polluted water. The factory owner bought some of the land, but the farmers lost their livelihoods and were also not compensated for their losses. Thus, farmers were driven to non-agrarian livelihoods. Others left their land fallow and sought occupations in Ramoji Film City, as the income is higher and the livelihood seems to be less volatile. A second conflict is more clandestine.

The amount and quality of village's largest reservoir has been reduced, as the Ramoji Film City diverts water from its feeder creek. It is still used as source for domestic water and for livelihood activities like fishing, if the amount of water rises sufficiently after the monsoon.

Dairy farming saw a decrease in the last 20 years, but those who remained in the business have increased and professionalized their business. During our visit, we also saw as new livelihood, a laundry service, catering to hotels and hospitals in Hyderabad.

The water supply to households was in a transformative stage during our visit. Under the "Mission Bhagiratha" Telangana state aims at providing 100 L of piped treated drinking water to households in rural areas [118]. In March 2019, the pipelines were being laid in the village, but there was no water in the pipes yet. Thus, the population was still relying on the supply from the gram panchayat, sourced from the local tank. Each household got two hours of water supply on every second day. The water has to be stored by each household individually, so one finds plastic barrels and other storage facilities in front of almost every house. Parallel to that, there is a drinking water supply, delivering treated water from river Krishna to public standposts. Water is available here between one to two hours daily. Thus, for households two parallel water supply infrastructures are in place and a third is in the making (Figure 4).

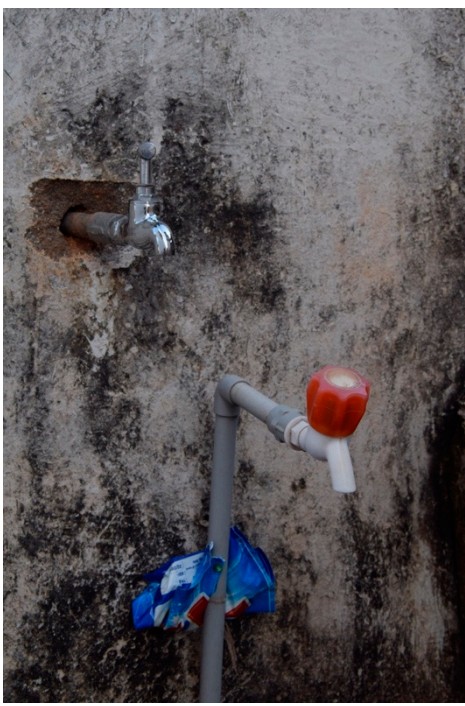

**Figure 4.** Two parallel supply systems—two water taps per household (photograph by C. Butsch).

Earlier, the water supply was mainly from ground water sources. Supply from the tank was initiated by the local government when the ground water got contaminated by the polluting industry in the 1980s. The Krishna drinking water supply project was initiated by the state government in 2005. However, most people in Anajpur use for drinking and cooking RO water, which is delivered from a plant in the neighboring village.

In Anajpur, the hydrosocial cycle has seen significant changes, which came along with conflicts. New, powerful actors detached the traditional water users from their water sources. While the population and the local government could develop some coping mechanisms, this still contributed to changes of the village's occupational structure, meant economic loss for several households and resulted in new water infrastructures and water related practices.

### 3.4. Periurban Hyderabad—Bowrampet

Bowrampet is situated just inside the area enclosed by the ORR. It is well connected to the centre of the agglomeration, resulting in increased transformation. In 2011, the census counted 5317 inhabitants, but during our visit, signs of recent and massive growth were visible leading to the assumption that the village population is much higher in 2019. New large-scale settlements came up outside the village core, few as apartment complexes, more as luxurious single homes.

Bowrampet has been severely affected by droughts in the last years prior to our visit. Respondents describe a decline of available water Surface water was until the 1980s the primary source of water for the village. Three larger artificial storages and two smaller ones were sufficient to secure the water for livelihoods like farming and fishing. In the 1980s, the villagers shifted to groundwater as main source of water, as it was initially available round the year and because the demand for water was rising (new cropping patterns, increasing population). The groundwater level has fallen quickly from 150 feet in the 1980s to 1000 feet (only these deep borewells provide water constantly). Older farmers said that the groundwater level fell more quickly after 2006, due to the development initiated in this area by the ORR project (finalized in 2010). But the groundwater is not only used locally: we observed a lot of water tankers delivering water to the city—and groundwater fed filling stations for them (Figure 5). Thus, the Bowrampet's groundwater has been commodified and became an export good, which is more remunerative than selling vegetables or fish.

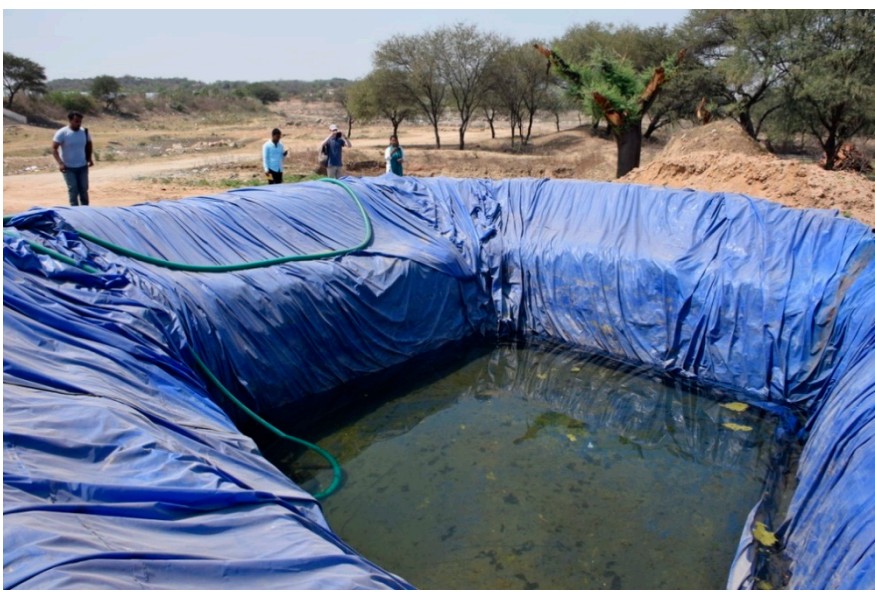

**Figure 5.** Sump of a water trader (photograph by C. Butsch).

Tanks surrounding the village have been desilted and restored with public funds. Yet, because of the drought in the last years, the tanks did not carry enough water for fishing activities and also could hardly be used for traditional irrigation. Additionally, former prime irrigation land, just below the village's largest tank has been transferred into a residential complex. All over the village, land-owning farmers sold their land, which increased in value quickly, with the implementation of the ORR, turning agriculture into a side business.

Other water-based livelihoods are influenced by periurban transformation, too, e.g., pottery. One potter told us, that he still follows the traditional practices, but has upscaled his business. Like the other potters of Bowrampet he now delivers his products to urban markets in Hyderabad. Input materials are not sourced locally anymore. New water-based livelihoods that emerged, because of the increasing entanglements are dairy farming and water vending.

The water supply in the village is highly commodified. After the panchayat set up a RO-plant in 2009, private providers followed suit and almost all villagers now rely on RO water for drinking, which is supplied by an estimated 20 RO plants, out of which the majority is not formally registered. This from of water supply brings completely new practices, which evolve around picking up water, storing water etc. For domestic water supply, there is a distribution system in place, which is groundwater fed and delivers to each household every second day water for one hour (less in summertime), which then is stored individually. In addition, some households also use groundwater for their domestic consumption.

To conclude, the hydrosocial cycle has seen fundamental changes. The village's hydrology has changed completely due to a mix of anthropogenic and climatic factors. Water scarcity has speeded up the process of farmers quitting agriculture. In different areas water has become a commodity—as drinking water but also as an alternative product for farmers to sell for the city. Through periurbanisation all water-based livelihoods have changed drastically.

### 3.5. Periurban Kolkata—Hadia

Hadia is a group of smaller hamlets located in the East Kolkata Wetlands (EKW), an area protected under the Ramsar convention. The wetlands are a prime example of a man-made waterscape, as all the water flows in this system of ponds, creeks and canals are following artificially created water courses. One central function of the wetlands is to treat Kolkata's sewerage, which is used here since colonial times as an input for farming and aquaculture. Due to protection status development is restricted preventing e.g., large scale housing projects.

Hadia had 7921 inhabitants in 2011 [119]. Almost all of its inhabitants belong to scheduled castes, which is related to the village's wastewater-based livelihoods, regarded as ritually impure. The hamlets are located north and south of highway number 3, which was built parallel to a large waste water canal, cut by the colonial administration. The wastewater is distributed among the villages in the EKW through mutual agreements.

North of the highway, in the hamlet Bamanghata north, several medium sized ponds, called *bheris* in the local language Bengal, are used for intensive aquaculture by individual fish farm owners. These small *bheris* have been created since the 1990s. After an initial investment in landscaping (creation of bunds and connecting the pond to the waste water canal) and fingerlings, the income is much higher than from agriculture.

One fishing cooperative in the village has 146 members (Figure 6). They own three large tanks which are farmed in a shifting system, so that the members can fish the whole year round. These tanks earlier belonged to the local land lord, the *Zamindar*. In 1980 the government of West Bengal sold the land to the fishermen and encouraged them to set up a cooperative. The cooperative receives subsidies from the government, e.g., for desilting the tanks regularly, in form of free fingerlings and recently by financial aids to set up infrastructure to attract local tourists. Work in the cooperative brings the fishermen a small but stable income and most of them follow other occupations as daily labourers in the tanning industry or in the city.

South of the highway in Bamanghata South and Bagdoba, the landscape looks very different, as it is slightly elevated and water does not flow directly here from the main canal. Thus, one finds many small, rainfed pods, which are also used for aquaculture, mainly for subsistence production. The population is engaged in waste-water-based agriculture and also works outside the village as daily wage labourers.

Household water supply in Hadia is mainly secured by individual handpumps in each house. As the groundwater is with roughly 150 feet easily accessible. There is a severe issue regarding the quality of groundwater (e.g., Iron, Arsenic). Therefore, the gram panchayat started to drill tube wells, which provide water from a depth of approximately 1000 feet. Yet, this water is also believed to be contaminated and is thus used mainly for

domestic purposes. For drinking water most villagers rely on RO water, which is delivered by local water vendors.

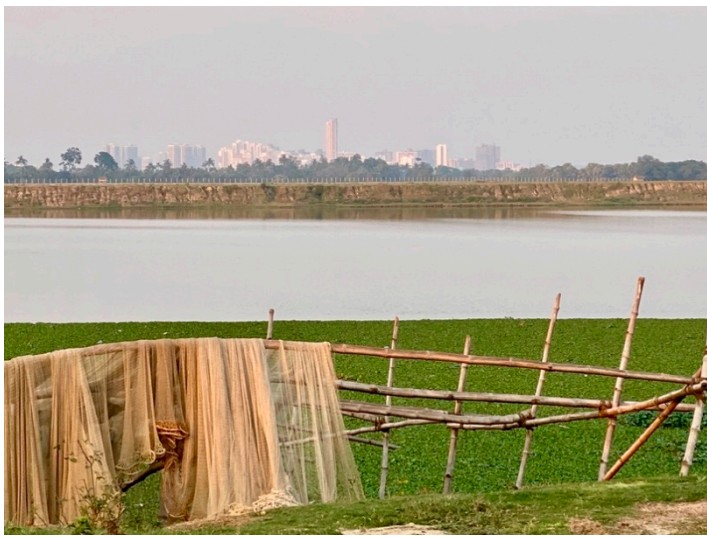

**Figure 6.** View of the largest tank of the fishing co-operative with a vista of Newtown, a satellite city of Kolkata (photograph by C. Butsch).

Hadia's hydrosocial cycle is closely related to the village's periurban situatedness, as it the main water input for this region stems from Kolkata. As the scope for transformation is very limited, due of the protection status, the waterscape is likely to remain relatively stable. Changes observed in the past are related mainly to changing governance regimes—with the transfer of land from the big land owners during colonial times to small holders in successive reforms after independence. This resulted in an increase of small aquacultures and an increasing demand for waste-water as valuable input.

*3.6. Periurban Kolkata—Badai/Talbanda*

Badai, is part of a group of villages, called Talbanda, located in "Bilkanda II" gram panchayat. South of the Talbanda is a newly constructed highway. Close to the highway are clusters of dyeing industries and medium sized engineering workshops. Badai is located in the northern part of Talbanda, characterized by a less dense settlement structure and the absence of industries. In 2011 Badai had 3068 inhabitants, with a majority of the population being Muslims [120]. Badai is surrounded on three sides by agricultural land, which is intensively used for paddy and vegetables. There is a lot of common property, inter alia the mango orchard and the irrigation system, which was installed by the panchayat but is maintained collectively by farmers.

Land plots are relatively small and several farmers have additional incomes as unskilled labourers, shop keepers and the like. The good road connection to Kolkata since the 1980s encouraged villagers to seek additional income from urban occupations. In between the fields there are several individually owned ponds, which are used for aquaculture. In Badai itself farming and fishing are the main water-based livelihoods. Due to the intensive industrialization and urbanization in the southern part, agriculture is affected negatively. Two processes led to a large area of erstwhile fertile land becoming infertile: Dyeing industries from Talbanda release untreated effluents into the creeks and canals. One of the main canals, which used to bring irrigation water to the fields is now a blackish flow of sewerage. Some low-lying fields have been flooded by effluents directly making farming impossible. In addition, construction activities south of Badai resulting in the blocking of important drainages, resulting in water logging on low-lying fields. Thus, the traditional livelihoods are severely endangered by the emerging water-based livelihoods—mainly the industrial activities.

While visiting a dyeing factory, we got a first-hand impression of the rudimental construction: the building itself consisted of bamboo and metal sheets water was heated with a charcoal fired oven and machinery seemed very old. The factory was not part of vertically integrated company, but specialized in dyeing only. Talbanda is only the production site, but the closeness to the Kolkata metropolitan region makes this periurban village a good location. Additionally, the gap in enforcement of environmental laws by local authorities is an advantage. So far, there were only halfhearted attempts to install a central sewage treatment plant.

In Badai, there are three different sources of drinking water: some households still rely on the traditional hand pumps, others get their water from public stand posts and some buy RO water. A large, government certified bottling plant sells 20-L jars, which are delivered by small, informal vendors, who buy the water from the plant and earn their living from the delivery (Figure 7). Piped water is provided to the stand posts by the Public Health Engineering Department of the West Bengal Government. Untreated groundwater is available here three times a day for one and a half hours. Especially in the summer months respondents reported shortages of domestic water, mainly for those who rely on their own hand pumps, because of ground water depletion.

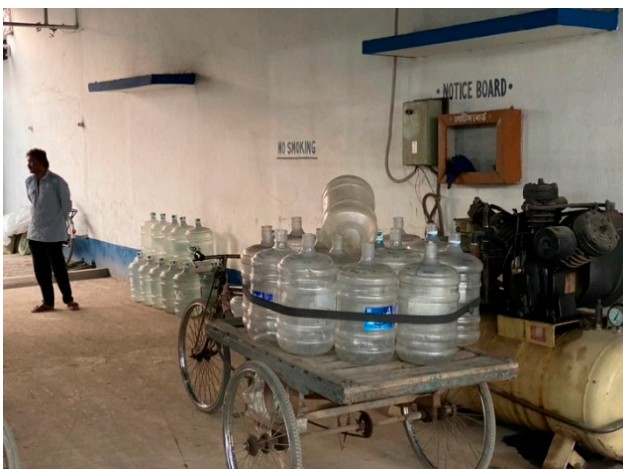

**Figure 7.** A water-vendor collecting jars at the licensed RO-plant (photograph by C. Butsch).

## 4. Discussion

The tales from the six villages first and foremost illustrate that the periurban is a heterogeneous space, a zone in transition, where the final stage is not necessarily predetermined. While implicitly it is often assumed that the peri-urban may eventually become urban, we would like to argue, that the futures of these spaces in the making are still open and can be influenced—and directed in a sustainable direction, if the right measures are taken.

Related to the transitional nature of the periurban is the emergence of new actors, e.g., developers or industries. With them, new water demands emerge, putting the resource under stress and resulting in a re-making of periurban waterscapes. In this re-making, new actors may play out their power to deprive traditional users, like fishermen or farmers, from their sources of livelihoods—willingly or as a side effect, for example if industries pollute waterbodies while saving the money of sewage treatment. But also growing cities themselves—with their demand for fresh water—become a powerful actor re-making the waterscape to ensure that their thirst is satisfied.

The application of the hydrosocial cycle perspective contributes to the understanding of how peri-urban areas emerge as a product of physical-social coproduction. Reading the case studies in the six tales through this specific political ecology lens adds to the understanding of waterscapes as hybrids of the societal and the natural sphere. Water availability and water quality in the case studies are the result of a man-made landscape as

a manifestation of political will and power. But the hydrosocial perspective also allows for the necessary process-based understanding of the periurban transformation as it addresses the changing relation between water and society during the periurban transformation. One of these changes is the commodification of water. Farmers start to sell water instead of rice and vegetables and the distinction between drinking water (with RO-water emerging as a new standard, also through social practices) and other "waters" (household water, agricultural water etc.) increases the complexity of periurban waterscapes. The three perspectives applied in the analysis—institutions, livelihoods, water as a consumption good—allow for a differentiated understanding of the genesis of the waterscapes and the ongoing changes of the hydrosocial cycle.

From an institutions perspective, the six tales illustrate the important, yet very different impact of policies. In Uruli Kanchan the cutting of canals first led to an increase in fresh water and recently the diversion of the canal's water to the upstream city to decrease in water quality—both affecting farmers' livelihoods. But also in the other villages subsidies shape cropping patterns. Alterations in the waterscape are part of this, as the provision of irrigation facilities in forms of bunds, pumps, deep tubewells canals etc. are important for means of securing voters for local politicians. In the two case studies involving fishing, one (Hadia) illustrates the supportive power of politics to turn fishing into a sustainable livelihood, while in the other (Paud) policies result in traditional communities being deprived from their traditional livelihood. The Hadia case study further illustrates the potential power of rules and regulations in shaping the future of the periurban. Due to the protected status under the Ramsar convention, EKW is likely to sustain its ecological function in future. But the periurban transformation also creates new water rights: When a village becomes part of a municipal corporation, the entitlement to water changes over night. And the example of the township erected in Paud shows, that even within the same jurisdiction water rights can be very different: while the inhabitants of the township enjoy 24/7 water supply, the inhabitants of the "old" village have only access to water for half an hour daily.

Livelihoods in the periurban areas of the three agglomerations are under the pressure to either intensify and/or professionalize or to quit. This is not in all cases exclusively related to the alterations of the hydrosocial cycle, but there are strong interdependencies. Intensification can be observed in the case of Bowrampet's potters, who now produce large scale for urban markets, or Anajpur's remaining dairy farmers. Bowrampet's farmers quit their traditional livelihoods to sell water while Bowrampet's fishermen do not find enough water in their ponds to continue this traditional livelihood. In other cases, conflicts related to alterations of the waterscape result in traditional users being deprived from their livelihoods, e.g., when water is diverted (Anajpur, Uruli Kanchan) or polluted (Anajpur, Badai). These conflicts are also related to the fragmented, unclear, or overlapping rules in periurban areas. At the same time new livelihoods emerge in the periurban, partly because of new demands (car wash, laundry services) but also, because of a changing perception of water itself (water vending). In some cases new livelihoods pollute water in a way, that makes it unusable for others, e.g., in the case of dyeing industries. Thus, some water-based livelihoods have a strong negative impact on the hydrosocial cycle.

The changing relation between water and society becomes very obvious from the analytical perspective of water as a consumption good, which results in a making of several new water infrastructures. The increasing demand observed in all six villages gives birth to various "socio-physical constructions" [20]: borewells, pipelines, canals, RO-plants etc., which often are transient in nature. In addition, in most of cases a multiplicity of sources emerges at the same time. The state's mission to provide safe and healthy drinking water together with the pollution of water bodies—above and below ground—and the changing aspirations of consumers result in a confusing variety on the supply side. RO-water is becoming a norm in almost all of our study sites. It is actively promoted by several actors, often with good reasons. Yet, it can be assumed that this multiplicity of water

supply is only a temporary phenomenon, related to the transitional stage of the periurban hydrosocial cycle.

## 5. Conclusions

India's current urbanization process results in the emergence of very specific zones of transition around agglomerations. This process is—as the current urbanisation in Asia and Africa as a whole—not comparable with the urbanisation that occurred in other regions of the world in the past. Thus, new perspectives and theories are needed. The application of a hydrosocial cycle perspective opens new views on periurban interfaces as emerging mosaic of unique waterscapes– often existing only temporarily. The meaning of water, the rights to access water and the water related infrastructure are constantly renegotiated, as permanently new water demands emerge and new actors enter the scene. Especially this process-based understanding links the theoretical lens of the hydrosocial cycle with the object of investigation, the periurban space.

The findings presented above indicate, that the relation between water and society is changing in several ways and so does the way it is governed. Drinking water for example is subject to commodification processes, which is related to changing concepts of health and hygiene on a societal level that translate into the emergence of new value chains of producing drinking water. Also, for farmers the meaning of water is changing as they can either sell it directly as a good, instead of using it for crops, or start using waste water for their farming activities. The tales from the six villages further illustrate that the emergence of new actors and new activities in periurban India result in power struggles over the increasingly sparse resource. These take place in a setting, where old institutions are eroding and new ones are emerging, resulting in a high degree of informality. One negative outcome of this constellation are unsustainable water practices, like the overextraction of groundwater, which potentially contributes to the vulnerability of less powerful actors like marginal farmers.

Going beyond our first probing of periurban hydrosocial cycles, a deeper understanding of the periurban specifics of water as a basis for livelihoods and a consumption good and the rules and regulations shaping these waterscapes is needed. This will allow for actively shaping the periurban transformation and eventually steer it in a more sustainable direction. One way of doing so can be action oriented transformative research, aiming at involving local actors in the development of alternative transformative pathways towards a sustainable future [121–123]. Designing these pathways with local communities is the goal of the next phase in the authors' joint research project. These pathways will enable communities to identify common goals and empower them to develop strategies to leave the often unsustainable "business-as-usual" scenario. As periurban futures are not pre-determined there is ample space for shaping them in more desirable ways.

**Author Contributions:** Conceptualization, C.B.; methodology, All Authors; formal analysis, All Authors; writing—original draft preparation, C.B., S.C., S.L.G.; writing—review and editing, S.K., L.M.H.; visualization, C.B.; project administration, C.B., S.C., L.M.H.; funding acquisition, C.B., L.M.H. All authors have read and agreed to the published version of the manuscript.

**Funding:** The project "H$_2$O-T2S in Urban Fringe Areas" is financially supported by the Belmont Forum and NORFACE Joint Research Programme on Transformations to Sustainability, which is co-funded by AKA, ANR, DLR/BMBF, ESRC, FAPESP, FNRS, FWO, ISSC, JST, NSF, NWO, RCN, VR, and the European Commission through Horizon 2020 under grant agreement No. 730211.

**Informed Consent Statement:** Informed consent was obtained from all subjects involved in the study.

**Data Availability Statement:** The data are not publicly available due to data protection of the interviewees.

**Acknowledgments:** Sincere thanks to Kranti Yardi, Partha Banerjee, Kanak Umredkar, Rushikesh Dhumal, Vaishnavi Uchagonkar Ranjit Guha, Basudev Baneerjee and Sai Kiran for translation and logistical support during the field visit in March 2019. We would like to thank three anonymous reviewers for the encouraging and valuable feedback on the first version of the manuscript.

**Conflicts of Interest:** The authors declare no conflict of interest.

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
