# Peer review of "Changing Hydrosocial Cycles in Periurban India"

_land, doi:10.3390/land10030263_

Round 1

Reviewer 1 Report

This is an interesting paper on a topic of key relevance to sustainable development research in Indian and beyond. The analysis is thorough and the theoretical framework impressive. If the paper lacks anything, it is some further reflection in the conclusion on how the various complexities of the 'hydrosocial cycle' identified by the authors in the peri-urban settings under review (e.g., power struggles, poor governance and unsustainable water practices, etc.) might be resolved in the future. I would, therefore, suggest some minor revisions of the conclusion are required before publication. 

Author Response

Thank you for this encouraging review. We rephrased the conclusions, hoping that they now add better to the overall message of the article.

Reviewer 2 Report

The paper addresses an interesting topic, still there are several issues that need to be strengthened and/or corrected on the paper, namely:

Literature review misses several seminal works and important advances crossing land use change and waterscape planning...

Material and methods are hard to understand because the research steps are not adequately described.

Conclusions need to be more scientific. They are too broad and general highlight the limitations of the research and they appear to be very too empirical for a research paper.

Author Response

The paper addresses an interesting topic, still there are several issues that need to be strengthened and/or corrected on the paper, namely:

Literature review misses several seminal works and important advances crossing land use change and waterscape planning...

Response: Thank you for bringing this to our attention. Given the comment by reviewer 3 that our literature list is  – according to her/him – is too long, we now added one more reference addressing this topic, which we find to be one which has a high number of citations (Eakin/Lerner/Murtinho 2010) and a more recent paper by the same authors (Lerner et al. 2018). We hope to have addressed this point adequately or would kindliy like to request a more specific hint, which authors or concrete publications to include.

Material and methods are hard to understand because the research steps are not adequately described.

Response: Thank you for this comment. This section was reorganized and further information was added, hoping that this part is clear now.

Conclusions need to be more scientific. They are too broad and general highlight the limitations of the research and they appear to be very too empirical for a research paper.

Response: Thank you for this comment. Thank you for this comment. We rephrased the conclusions, hoping that they now add better to the overall message of the article

Reviewer 3 Report

  1. The research methodology is only superficially explained and does not allow to understand which research methods were used and what they were used for, which limits the understanding of the results of the research work carried out.
  2. The work is carried out based on qualitative research, but the sample covered in the research is not mentioned, nor is reference to the data collection instruments used.
  3. The work would have greater scientific value if it combined qualitative and quantitative methods, especially to interpret how the relationships of the water resource with society are related to other factors such as policies for the development of infrastructure and housing.
  4. There are many parts of the text written in the first person, which constitutes an error in scientific writing, please check.
  5. The graphs, photos and diagrams are not properly cited and it is not known which ones are obtained by the authors as part of the research work.
  6. The conclusion of the work does not respond to the objective stated on page 2. To demonstrate, based on exploratory field work in six villages, how the relation of water and society is changing in periurban India. Please check.
  7. For a field research work, a bibliographic reference list of 120 documents seems to be too long
  8. Some comments have been made in the article using the adobe acrobat reader tool.

Author Response

The research methodology is only superficially explained and does not allow to understand which research methods were used and what they were used for, which limits the understanding of the results of the research work carried out.

Response: Thank you for this comment. This section was reorganized and further information was added, hoping that this part is clear now.

The work is carried out based on qualitative research, but the sample covered in the research is not mentioned, nor is reference to the data collection instruments used.

Response: Thank you for this comment. The sample size was added for each of the three methods applied during the pilot study.

The work would have greater scientific value if it combined qualitative and quantitative methods, especially to interpret how the relationships of the water resource with society are related to other factors such as policies for the development of infrastructure and housing.

Response: It would of course be good to approach the topic with a mixed-methods-research approach and this is part of the project’s second phase. However here we can only present the data at hand and hope that this is sufficient.

There are many parts of the text written in the first person, which constitutes an error in scientific writing, please check.

Response: Thank you for this comment. We have made the necessary changes.

The graphs, photos and diagrams are not properly cited and it is not known which ones are obtained by the authors as part of the research work.

Response: Thank you, indeed this was missing and we have added the sources now.  

The conclusion of the work does not respond to the objective stated on page 2. To demonstrate, based on exploratory field work in six villages, how the relation of water and society is changing in periurban India. Please check.

Response: Thank you for this comment. We rephrased the conclusions, hoping that they now add better to the overall message of the article.

For a field research work, a bibliographic reference list of 120 documents seems to be too long

Response: Thank you for this comment. As we would like to link our empirical work to the ongoing theoretical discourses of the disciplines in which the authors involved are rooted, the list of references is quite extensive, but we hope that this is acceptable, especially with a view to the digital format of the publication and regarding the fact that reviewer one even requested further references.

Some comments have been made in the article using the adobe acrobat reader tool.

Response: Thank you for providing these detailed markups. We have changed the text accordingly, Just where you requested a change in line 89 “Swyngedouw” we did not make a change, as this is the author’s name we are quoting here.